# Cholesterol-Lowering Mechanism of *Lactobacillus* Bile Salt Hydrolase Through Regulation of *Bifidobacterium pseudolongum* in the Gut Microbiota

**DOI:** 10.3390/nu17183019

**Published:** 2025-09-22

**Authors:** Yingying Liu, Weijia Kuang, Man Li, Zhihao Wang, Yanrong Liu, Menghuan Zhao, Hailin Huan, Yao Yang

**Affiliations:** 1School of Food Science and Pharmaceutical Engineering, Nanjing Normal University, Nanjing 210023, China; liuyingying010810@163.com (Y.L.); 17314410523@163.com (W.K.); liman411521@163.com (M.L.); wang1001zhihao@163.com (Z.W.); 15062138603@163.com (M.Z.); 2Nanjing Institute of Product Quality Inspection, Nanjing 210019, China; lyr090@163.com; 3Jiangsu Academy of Agricultural Sciences, Nanjing 210023, China

**Keywords:** bile salt hydrolase, cholesterol-lowering, *Bifidobacterium pseudolongum*, gut microbiota, farnesoid X receptor

## Abstract

**Background:** Cardiovascular diseases (CVDs) represent a major global health burden, and cholesterol reduction is a key strategy for their prevention and management. This study investigated the mechanism by which bile salt hydrolase (BSH) from *Lactobacilli* reduces cholesterol levels by modulating the growth of *Bifidobacterium pseudolongum*. **Methods:** The BSH-recombinant strain YB334 was administered to high-cholesterol-diet mice, and the cholesterol-lowering function of the strain was evaluated by assessing serum cholesterol parameters, including total cholesterol (TC), low-density lipoprotein (LDL) and high-density lipoprotein (HDL). Metagenomic sequencing was used to analyze the gut microbiota, leading to the screening and acquisition of the “responsive” strains affected by BSH. Subsequent investigations were conducted into their cholesterol-lowering effects and mechanisms of action. **Results:** Oral administration of the BSH-recombinant strain YB334 can effectively reduce serum cholesterol levels in hypercholesterolemic mice while simultaneously leading to a significant increase in the abundance of *B. pseudolongum* within the gut microbiota. In vitro experiments indicated that this increased abundance might be closely associated with the strain’s high tolerance to CA, the catalytic product of the BSH enzyme. The BPL-4 strain, obtained through screening, demonstrated cholesterol-lowering efficacy. Mechanistically, BPL-4 altered bile acid pool composition and modulated the farnesoid X receptor (FXR) signaling axis: it suppressed ileal FXR-fibroblast growth factor 15 (FGF15) expression, thereby de-repressing hepatic cholesterol 7α-hydroxylase (CYP7A1) and accelerating cholesterol catabolism into bile acids. **Conclusions:** This study provides the first evidence that BSH from lactobacilli can shape the signature gut microbiota by modulating bile acid metabolism via the FXR-CYP7A1 axis, thereby demonstrating a mechanism for its cholesterol-lowering effects.

## 1. Introduction

Cardiovascular diseases (CVDs) remain the leading cause of morbidity and mortality globally [1]. Reducing cholesterol levels is crucial to both prevention and management [2]. Although pharmacological agents (e.g., statins) are available, their use may cause adverse effects, prompting interest in effective, safe, and health-promoting alternatives for cholesterol reduction [3,4]. Probiotics with cholesterol-lowering properties have attracted increasing attention. For example, Yue Li et al. [5] reported that *Lactobacillus plantarum* H6 can significantly lower serum cholesterol levels in hypercholesterolemic mice by modulating intestinal microbiota composition (notably increasing *Muribaculaceae* abundance) and regulating microbial metabolites (including short-chain fatty acids and vitamins). Similarly, Narathip Puttarat et al. [6] validated that specific probiotics (such as *Limosilactobacillus reuteri* TF-7, *Enterococcus faecium* TF-18, and *Bifidobacterium animalis* TA-1) can markedly reduce serum cholesterol levels in hypercholesterolemic rats via bile salt hydrolase activity and cholesterol assimilation, with synergistic lipid-lowering effects observed in mixed-strain treatments. Despite these advances, the mechanisms by which probiotics lower cholesterol levels remain incompletely understood [7,8].

Bile salt hydrolase (BSH, EC 3.5.1.24) is prevalent in the mammalian gut microbiota, including *Lactobacillus*, *Clostridium*, *Bacteroides*, and *Bifidobacterium*. Among these, *Lactobacillus* and *Bifidobacterium* are the primary sources of high-yield BSH bacteria used in in vitro studies [9,10]. BSH catalyzes the hydrolysis of conjugated bile acids, releasing free bile acids and amino acids (glycine/taurine) [11,12,13]. Multiple studies have linked BSH activity to cholesterol-lowering effects. For instance, Fengjie Huang et al. [14] reported that theaflavins in Pu’er tea can inhibit BSH activity, elevate conjugated bile acids levels, obstruct the farnesoid X receptor-fibroblast growth factor 15 (FXR-FGF15) signaling pathway, and diminish cholesterol levels in both mice and humans. Similarly, Guangqiang Wang et al. [15] engineered a high-activity BSH strain which attenuated hepatic steatosis by enhancing bile acid deconjugation, suppressing the FXR signaling, and upregulating cholesterol 7α-hydroxylase (CYP7A1) expression. Our laboratory has conducted continuous research on the cholesterol-lowering properties of the lactic acid bacterium BSH [16]. Experiments have also shown that the BSH recombinant strain bacterium derived from YB334 exhibits cholesterol-lowering properties through FXR-mediated regulation of bile acids after TCA hydrolysis [17]. Recent investigations have documented this discovery, which is consistent with our findings. Joyce et al. [18] reported that *Escherichia coli* producing BSH from recombinant *Lactobacillus salivarius* exhibited probiotic effects, including lowering plasma cholesterol levels, managing body weight, and modulating glucose and lipid metabolism in mice fed a high-fat diet. The study findings indicated a notable reduction in hepatic FXR expression and a substantial increase in small intestine FXR expression. Conversely, Lina Yao et al. [19] documented that a *Bacteroides* strain with a disrupted BSH gene markedly lowered plasma cholesterol levels in mice on a high-fat diet, resulting in a significant decline in hepatic FXR expression without considerable alteration in small intestine FXR expression.

Previous studies from our laboratory have shown that when the BSH recombinant strain YB81 was administered to PGF mice with antibiotic-depleted gut microbiota, this strain did not exhibit cholesterol-lowering effects. However, in SPF mouse models with normal gut microbiota, the same strain demonstrated significant cholesterol-lowering effects. These results indicate that the gut microbiota directly contributes to the cholesterol-lowering effects of certain BSH from Lactobacilli [20].

Building on these findings, this study aims to elucidate the cholesterol-lowering mechanisms of the recombinant strain YB334 by analyzing BSH enzyme alterations in the structure and function of the gut microbiota. We seek to uncover novel mechanisms of cholesterol regulation by lactic acid bacterial BSH from the perspective of host-microbe interactions, thereby providing a theoretical foundation for developing probiotic therapies against hypercholesterolemia.

## 2. Materials and Methods

### 2.1. Experimental Strain and Culture

All strains used in this study are listed in Table 1. *Lactobacillus* and *Bifidobacterium* were cultivated anaerobically at 37 °C. *Lactobacillus* and *Limosilactobacillus* were cultured aerobically in de Man, Rogosa and Sharpe (MRS) medium, while *Bifidobacterium* was cultured anaerobically in modified MRS agar medium (supplemented with mupirocin lithium salt and cysteine hydrochloride) [21]. Both media were obtained from Haibo Ltd. (Qingdao, China). Erythromycin (Em^r^; Sangon Biotech Ltd., Shanghai, China) was added at a final concentration of 10 μg/mL.

### 2.2. Animal Experimentation

All mice were obtained from Gempharmatech Co., Ltd. (Nanjing, China). The animal study protocol was approved by the Animal Ethics Committee of Nanjing Normal University (Approval Number: SYXK 2020-0047; Approval Date: 15 February 2024), and all experiments were performed in accordance with relevant guidelines and regulations.

Six-week-old male C57BL/6 mice (specific pathogen-free, SPF grade) were used in this study. They were housed in SPF conditions at the Experimental Animal Centre of Nanjing Normal University under a 12 h light/dark cycle with temperature maintained at 23 ± 2 °C and relative humidity at 55 ± 5%. The normal diet (ND) and high-cholesterol diet (HCD; containing 15% fat, 1.25% cholesterol, and 0.5% sodium cholate) were purchased from Jiangsu Synergy Pharmaceutical Bioengineering Co., Ltd. (Nanjing, China).

Following one week of adaptive feeding, the mice were divided into two groups: the control group received an ND, while the experimental group was fed an HCD to induce hypercholesterolaemia. Six mice were allocated to each group. Thereafter, the experimental group received a daily oral gavage of 200 μL of a bacterial suspension (1 × 10^9^ CFU/mL). In contrast, the ND and HCD control groups were administered equivalent volumes of phosphate-buffered saline (PBS; Sangon Biotech) and skim milk (SM; Anchor, Auckland, New Zealand), respectively.

Preparation of bacterial suspensions:Recombinant lactic acid bacteria (NB5462 and YB334): Single colonies were inoculated into liquid medium and cultured overnight for 12 h. The cultures were then diluted in fresh MRS medium to an OD_600 nm_ of approximately 0.1, and incubated at 37 °C until the OD_600 nm_ reached around 0.3. Protein expression was induced by adding the SppIP inducer peptide (amino acid sequence: MAGNSSNFIHKIKQIFTHR; Genscript Biotechnology Co., Ltd., Nanjing, China) at a final concentration of 50 ng/mL. Cultivation continued until an OD_600 nm_ of 2.5 was achieved. The cells were harvested by centrifugation, washed, and resuspended in PBS for storage at 4 °C.*Bifidobacterium pseudolongum* BPL-4: A single colony was inoculated into liquid medium and cultured for 16 h overnight. A 1% (*v*/*v*) inoculum of this seed culture was transferred to 45 mL of fresh, modified MRS broth supplemented with mupirocin lithium salt and cysteine hydrochloride. After 12 h of incubation, the bacterial cells were collected by centrifugation. The pellet was resuspended in 10% sterile SM and stored at 4 °C.

The experimental period lasted for six to seven weeks, during which mouse body weight was monitored weekly. Upon completion of the gavage period, mice were fasted for 12 h (ad libitum access to water was maintained) and subsequently anesthetized with diethyl ether. Blood samples were collected via retro-orbital bleeding. Thereafter, the mice were euthanized by cervical dislocation. Immediately following euthanasia, the liver, ileum, ileal contents, and duodenum were aseptically collected, placed in sterile, nuclease-free microcentrifuge tubes, and snap-frozen at −80 °C for subsequent analysis.

### 2.3. Serum Biochemical Parameter Analysis

Whole blood was collected from mice via retro-orbital bleeding. After incubation at room temperature for 30 min, the samples were centrifuged at 1500× *g* for 15 min to isolate serum. The serum levels of total cholesterol (TC), triglycerides (TG), high-density lipoprotein cholesterol (HDL-C) and low-density lipoprotein cholesterol (LDL-C) were quantified using an automated biochemistry analyzer (Hitachi 7100; Hitachi High-Tech Corporation, Tokyo, Japan) at the Animal Centre of Nanjing Medical University.

### 2.4. Liver Histological Analysis

Liver tissues were fixed in 4% paraformaldehyde for 24 h, embedded in paraffin, and sectioned at a thickness of 4 μm. Sections were stained with hematoxylin and eosin (H&E) and imaged using an upright optical microscope (Nikon Eclipse E100, Tokyo, Japan). All histological procedures were performed by Wuhan Servicebio Technology Co., Ltd. (Wuhan, China).

### 2.5. Metagenomics Analysis

Metagenomic DNA was extracted from mouse feces using the E.Z.N.A.^®^ Soil DNA Kit (Omega Bio-tek, Norcross, GA, USA). Sequencing was performed on an Illumina NovaSeq/HiSeq Xten platform (Illumina, San Diego, CA, USA) following bridge PCR amplification. The remaining high-quality reads were de novo assembled using MEGAHIT (v1.2.9), with contigs ≥ 300 bp retained for subsequent analysis. Non-redundant gene catalogs were constructed from the assembled contigs using MetaGene for prediction and CD-HIT for clustering. All laboratory procedures and sequencing were conducted by Shanghai Majorbio Bio-Pharm Technology Co., Ltd. (Shanghai, China), and the bioinformatic analysis was performed on the Majorbio Cloud Platform (https://cloud.majorbio.com; accessed on 10 June 2024).

To identify microbial taxa with significant abundance differences between groups, linear discriminant analysis effect size (LEfSe) was employed, with a linear discriminant analysis (LDA) score threshold of >2. A custom reference database of bile salt hydrolase (BSH) genes was compiled by querying the UniProt protein database. Beta-diversity between experimental groups was assessed via principal coordinates analysis (PCoA) based on Bray–Curtis distances. Statistical significance of inter-group differences was determined using the Wilcoxon rank-sum test, with *p*-values adjusted for false discovery rate (FDR).

### 2.6. Isolation and Identification of B. pseudolongum in Mouse Feces

Fecal samples were obtained from mice in the HCD + YB334 cohort and suspended in PBS. The samples were then serially diluted and plated onto modified MRS agar supplemented with mupirocin lithium salt and cysteine hydrochloride. Following incubation, bacterial colonies were isolated. Genomic DNA was extracted, and the 16S rRNA gene was amplified using the universal bacterial primers 27F (5′-AGAGTTTGATCCTGGCTCAG-3′) and 1492R (5′-TACGGCTACCTTGTTACGACTT-3′). The polymerase chain reaction (PCR) products were purified and sent to Sangon Biotech for Sanger sequencing. The resulting sequences were analyzed using the basic local alignment search tool (BLAST) to identify homologous bacterial sequences (http://www.ncbi.nlm.nih.gov/BLAST; accessed on 5 October 2024).

### 2.7. BSH Activity Assays

The strain’s bile salt-degrading activity was assessed using the indophenol method [22]. An overnight culture was centrifuged, and the pellet was resuspended in 0.1 M HAc-NaAc buffer (pH 5.0; Sangon Biotech) to a final OD_600 nm_ of 5.0. This suspension was then mixed with a bile salt substrate containing 5 mM each of glycocholic acid (GCA) and taurocholic acid (TCA) sodium salts (Sigma-Aldrich, St. Louis, MO, USA). After incubation at 37 °C for 1 h, the enzymatic reaction was terminated by adding 100 μL of 15% trichloroacetic acid (Sangon Biotech). Subsequently, 100 μL of the supernatant was transferred, mixed with 900 μL of indophenol solution (Sangon Biotech), and vortexed thoroughly. The mixture was incubated in a water bath for 14 min, cooled in ice water, and its optical density was measured at 570 nm after standing for 5 min.

For the analysis of bile acids in murine fecal samples, Liquid chromatography-tandem mass spectrometry (LC-MS/MS) was performed using an ExionLC AD liquid chromatography system coupled to a QTRAP^®^ 6500+ mass spectrometer (Sciex; Framingham, MA, USA).

### 2.8. Effect of Addition of Crude Extract of YB334-BSH Enzyme on the Growth of Different Strains of Bacteria

Strain preparation was conducted as previously outlined for animal experiments, with an uninduced control group established and cultivated for 20 h. The bacterial culture was centrifuged at 8000× *g* at 4 °C for 15 min, rinsed with PBS, and resuspended in fresh MRS medium supplemented with TCA at a final concentration of 5 mM in ratios of 10:1 and 5:1 (*v*/*v*). The suspension was homogenized using glass beads (speed 6.00 m/s, cycle 10, 30 s on, 30 s off), followed by centrifugation at 12,000× *g* at 4 °C for 10 min. The supernatant was then removed, filtered through a 0.22 μm membrane filter, and preserved for future use.

Various test strains (BPL-1, BPL-2, BPL-4, BA-6, BL-4, BL-7, J5, J16, and L2) were initiated and cultured in their appropriate media until the logarithmic phase was reached. A 1% (*v*/*v*) inoculum was transferred to 200 μL of MRS + TCA medium containing a crude BSH enzyme extract derived from the YB334 recombinant strain, with MRS + TCA medium used as the control. The 96-well plate was positioned in a microplate reader (Varioskan LUX; Thermo Fisher Scientific, Waltham, MA, USA) and maintained at a constant temperature of 37 °C. The OD at 600 nm was recorded at 30 min intervals, and plate colony counts were performed after 5 h of incubation.

### 2.9. TCA/CA Bile Salt Tolerance

Two distinct bile salts, taurocholate acid sodium salt (TCA) and cholic acid (CA), were obtained from Sigma-Aldrich to evaluate the bile salt tolerance of the test strains [23]. Various strains were first inoculated onto appropriate solid media, and individual colonies were selected and transferred to corresponding liquid media for overnight incubation. The overnight culture were then inoculated at 1% (*v*/*v*) into fresh liquid media and incubated for 12 h. Subsequently, bacterial cells were harvested from three replicate cultures by centrifugation at 8000× *g* at 4 °C for 15 min. The pellets were washed and resuspended in equal volumes of either PBS buffer containing 5 mM TCA or PBS buffer containing 5 mM CA. The suspensions were incubated at 37 °C for 2 h, and viable counts (CFU/mL) were determined using the pour plate method to assess the survival of each strain in the presence of the respective bile salts.

### 2.10. RNA Isolation and RT-qPCR

Total RNA was isolated from mouse ileum and liver tissues using the RNAiso Plus Kit (Takara Biomedical Technology, Beijing, China). Following extraction, cDNA was synthesized using the PrimeScript™ RT Reagent Kit with gDNA Eraser (Takara Biomedical Technology, Beijing, China). Quantitative reverse transcription polymerase chain reaction (RT-qPCR) was then performed on a LightCycler System using TB Green Premix Ex Taq II (Takara Biomedical Technology, Beijing, China), with cDNA as the template. The amplification protocol followed the manufacturer’s instructions for the TB Green^®^ Premix Ex Taq™ II Reagent Kit on the LightCycler 480 System. Gene expression levels were calculated using the 2^−ΔΔCT^ method, with rpL32 mRNA as the internal control, and are presented relative to the HCD group. The primer sequences for RT-qPCR are listed in Table 2.

### 2.11. Statistical Analysis

All experimental data are presented as mean ± standard deviation (M ± SD). Graphical representation and statistical analysis were performed using GraphPad Prism software (version 9.5). Comparisons among multiple groups were conducted by one-way or two-way analysis of variance (ANOVA), followed by Tukey’s post hoc test for assessing significant differences between individual groups. A *p* value of less than 0.05 was considered statistically significant.

## 3. Results

### 3.1. BSH Recombinant Bacterium YB334 Has Cholesterol-Lowering Properties

Figure 1A,B illustrates that after 7 weeks of feeding an HCD, mice in the HCD group showed a substantial increase in body weight relative to those in the ND group (*p* < 0.001), confirming effective animal model establishment. Oral administration of either NB5462 of YB334 strains, delivered via gavage, exhibited a specific mitigating effect on weight gain in mice. Compared to the HCD + NB5462 group, the HCD + YB334 group exhibited a markedly smaller increase in body weight (*p* < 0.001), with no significant difference in body weight relative to the ND group (*p* ≥ 0.05). These results demonstrate that while a high-cholesterol diet can promote weight gain in mice, the recombinant BSH strain YB334 can more successfully mitigate this effect than the NB5462.

Figure 1C illustrates the serum cholesterol levels in mice. Compared to the ND group, TC levels were considerably elevated in mice in the HCD group (*p* < 0.001), validating the effective establishment of a hypercholesterolemic animal model. Compared to the HCD + NB5462 group, mice in the HCD + YB334 group exhibited a substantial decrease in TC levels (*p* < 0.001). Additionally, the HCD + YB334 group exhibited a substantial reduction in both LDL-C and HDL-C levels compared to the HCD group (*p* < 0.05), whereas the HCD + NB5462 group showed no significant change (*p* ≥ 0.05).

Figure 1D shows a histological study of the liver tissue stained with H&E. The HCD group displayed substantial microvesicular steatosis, characterized by abundant lipid droplet accumulation, which was absent in the ND group. All intervention groups exhibited varying degrees of attenuation in hepatic lipid deposition and vacuolation. The most significant amelioration was observed in the HCD + YB334 group, where hepatocyte morphology was largely restored to normal, resulting in superior outcomes compared to the HCD + NB5462 group.

These results demonstrate that the BSH recombinant bacterium YB334 can reduce TC, LDL-C, and HDL-C in hypercholesterolemic mice, thereby facilitating further investigation of the cholesterol-lowering mechanisms underlying BSH’s cholesterol-lowering effects.

### 3.2. BSH of Recombinant Bacterium YB334 Regulates Intestinal Flora Distribution

Metagenomic sequencing was performed to investigate intestinal microbiota alterations induced by the BSH recombinant bacterium YB334 (Figure 2). β-diversity analysis was employed to assess the variance among samples based on all non-redundant gene sets. PCoA (Figure 2A) revealed significant separation of the HCD, HCD + NB5462, and HCD + YB334 groups from the ND group, indicating substantial alterations in the gut microbiota structure among these three groups. Analysis of the genus-level intestinal microbiota in the four mouse groups (Figure 2B) revealed significant alterations in the relative abundances of 20 species, with the predominant genera being *Turicibacter*, *Ligilactobacillus*, and *Bifidobacterium*.

Subsequent analyses using LEfSe revealed that the microbial taxa exhibited substantial alterations in relative abundance among the groups (Figure 2C). Compared to the HCD group, the relative abundances of taxa such as *Eggerthellaceae*, *Lactiplantibacillus*, *Malacoplasma*, and *Erysipelatoclostridium* were enriched in the HCD + YB334 group. Notably, *Eggerthellaceae* has been linked to cholesterol metabolism [28], while *Erysipelatoclostridium* has demonstrated a negative correlation with cholesterol [29]. Compared with the HCD + NB5462 group, the relative abundances of genera including *Bifidobacterium*, *Limosilactobacillus*, *Erysipelotrichaceae*, and *Atopobiaceae* were augmented in the HCD + YB334 group. *Bifidobacterium* lowers cholesterol via BSH activity and modulating bile acid metabolism [30]; *Limosilactobacillus* can decrease TC levels [31]; and *Atopobiaceae* is a bacterium exhibiting extensive substrate specificity for BSH [32]. The relative abundance of *Erysipelotrichaceae* was significantly elevated in the HCD + YB334 group compared to the HCD and HCD + NB5462 groups. Collectively, these findings suggest that BSH generated by the recombinant bacterium YB334 can ameliorate HCD-induced dysbiosis, which is hypothesized to contribute to its cholesterol-lowering properties.

The HCD + YB334 group displayed the highest abundance of the dominant genus *Bifidobacterium*. Wilcoxon rank-sum testing (Figure 2D) revealed that the relative abundance of *B. pseudolongum* was markedly elevated in the HCD + YB334 group relative to the HCD + NB5462 (*p* < 0.05), with this increase being the most pronounced relative to the other gut microbiota. These findings suggest that the cholesterol-lowering effect of the recombinant strain YB334 is mediated through the enrichment of specific beneficial bacteria, notably *B. pseudolongum*, highlighting this bacterium as a key microbial target for elucidating the underlying mechanism.

### 3.3. Screening of B. pseudolongum BPL and Determination of BSH Activity

Preliminary metagenomic research revealed that the HCD + YB334 group possessed abundant gut microbial resources, with *B. pseudolongum* exhibiting the highest prevalence. Consequently, fecal samples from this group were used for isolating and screening *B. pseudolongum*.

Fresh feces were homogenized in sterile PBS, and serially diluted to 10^−5^, 10^−6^ and 10^−7^. A volume of 100 μL of the supernatant was plated on modified MRS agar supplemented with mupirocin lithium salt and cysteine hydrochloride, followed by anaerobic incubation at 37 °C for 48 h. The isolated single colonies were purified, and 66 colonies were selected for biochemical characterization and 16S rDNA sequencing. Subsequent analysis identified all the obtained strains as members of the genus *Bifidobacterium*. Further analysis using national center for biotechnology information (NCBI)-BLAST revealed 54 isolates as *B. pseudolongum*, subsequently named BPL. The remaining isolates were classified as two strains of *Bifidobacterium longum*, BL-4 and BL-7, and one strain of *Bifidobacterium adolescentis*, BA-6. In parallel, under aerobic conditions, *Lactobacillus johnsonii* J5 and J16 and *L. reuteri* L2 were isolated.

The BSH activities of the eight isolated strains of *B. pseudolongum* were assessed, and the results are presented in Table 3. The activity levels varied, leading to the selection of strains BPL-1 (moderate BSH activity), BPL-2 (low BSH activity), and BPL-4 (high BSH activity) for further experiments.

### 3.4. Effect of Crude Extract of Recombinant Bacterium YB334-BSH on the Growth of Different Strains

BPL-1, BPL-2, BPL-4, BA-6, BL-4, BL-7, J5, J16, and L2 were introduced into MRS + TCA medium supplemented with either inducible (+) or non-inducible (−BSH crude extracts derived from YB334 recombinant bacterial cultures and subsequently incubated for 5 h. MRS + TCA medium was used as a control. The experimental findings revealed three distinct scenarios:Figure 3A illustrates that, in contrast to the control group, the culture media infused with the YB334-BSH crude extract and inducer markedly enhanced the proliferation of many strains of *B. pseudolongum*, particularly when administered at a 10:1 supernatant-to-medium ration, demonstrating a significant growth-promoting impact, rising from 10^6^ CFU/mL to 10^7^ CFU/mL. The BPL-4 strain (accession No. PX061997) showed the most significant growth-promoting effects (Figure 3D). Conversely, the uninduced extract did not significantly influence the proliferation of *B. pseudolongum*;AS shown in Figure 3B, crude extracts of YB334-BSH, whether induced or non-induced, did not significantly enhance the development of *B. adolescentis* BA-6 and *B. longum* BL-4 and BL-7;As shown in Figure 3C, the induced YB334-BSH crude extract showed inhibitory effects on *L. johnsonii* J5 and J16, as well as *L. reuteri* L2, decreasing their counts from 10^7^ CFU/mL to 10^6^ CFU/mL.

The results revealed that the culture medium containing the induced YB334 crude extract demonstrated selective enhancement of growth across various bacterial strains, with the most pronounced effect observed in BPL strains.

### 3.5. Effect of Different Strains on Tolerance to TCA/CA Bile Salts

Upon simultaneous addition of TCA and CA to the culture medium at equal concentrations, all nine strains showed growth reductions compared to that in PBS (Figure 4). The experimental findings revealed three distinct scenarios:Figure 4A illustrates that the *B. pseudolongum* BPL-1, BPL-2, and BPL-4 displayed a notable tolerance to TCA, with a decline from 10^9^ CFU/mL to 10^4^ CFU/mL; however, they exhibited superior tolerance to CA, with a decrease from 10^9^ CFU/mL to 10^7^ CFU/mL;As shown in Figure 4B, *B. adolescentis* BA-6 was highly sensitive to TCA, declining from 10^10^ CFU/mL to 10^1^ CFU/mL, while it exhibited relatively good tolerance to CA, decreasing from 10^10^ CFU/mL to 10^7^ CFU/mL. Conversely, *B. longum* BL-4 and BL-7 demonstrated poor tolerance to CA, diminishing from 10^10^ CFU/mL to 10^1^ CFU/mL, but moderate tolerance to TCA, decreasing from 10^10^ CFU/mL to 10^8^ CFU/mL;Figure 4C illustrates that *L. johnsonii* J5 and J16, along with *L. reuteri* L2, demonstrate a notable resistance to both TCA and CA. TCA concentrations diminished from 10^8^ CFU/mL to 10^7^ CFU/mL, from 10^8^ CFU/mL to 10^7^ CFU/mL, and from 10^10^ CFU/mL to 10^9^ CFU/mL, respectively. CA diminished from 10^8^ CFU/mL to 10^6^ CFU/mL, from 10^8^ CFU/mL to 10^6^ CFU/mL, and from 10^10^ CFU/mL to 10^9^ CFU/mL, respectively. They also demonstrated the highest tolerance to TCA.

These data demonstrated that the BSH enzyme influences bile tolerance in a strain-dependent manner. Consequently, we hypothesized that such variations may result from the catalytic characteristics of the BSH enzyme, which catalyzes TCA substrate to CA, and that various strains may exhibit differing tolerances to TCA and CA.

### 3.6. B. pseudolongum BPL-4 Has Cholesterol-Lowering Effects

Figure 5A,B illustrates that after 6 weeks of an HCD, mice in the HCD group gained considerable body weight relative to those in the ND group (*p* < 0.001), confirming successful model establishment. Compared to the HCD group, the body weight of mice in the HCD + BPL-4 group was substantially reduced (*p* < 0.001). As shown in Figure 5C, serum cholesterol levels in the HCD group were markedly increased relative to those in the ND group (*p* < 0.01), validating the effectiveness of the high-cholesterol animal model. In comparison with the HCD group, TC levels in the HCD + BPL-4 group were markedly diminished (*p* < 0.05), and both LDL-C and HDL-C levels were significantly reduced (*p* < 0.05). Figure 5D shows a histological study of the liver tissue stained with H&E. Compared to the HCD + BPL-4 group, mice on the HCD alone exhibited substantial lipid droplet accumulation and marked fatty degeneration in adipose tissue, pathologies that were absent in the ND group. This indicates that BPL-4 effectively alleviates HCD-induced hepatic steatosis.

These results revealed that *B. pseudolongum* BPL-4 effectively lowered serum cholesterol levels in hypercholesterolemic mice.

### 3.7. BSH Indirectly Regulates Intestinal FXR in B. pseudolongum BPL-4

Figure 6A,B display bile acid profiles in the experimental groups. The HCD + BPL-4 group demonstrated markedly reduced conjugated bile acids (*p* < 0.001) and substantially elevated unconjugated bile acids (*p* < 0.001) compared to the HCD group. Analysis of individual bile acids revealed significantly reduced (*p* < 0.01) TCA levels and elevated (*p* < 0.001) CA levels in the HCD + BPL-4 group relative to those in the HCD group. The alterations in bile acid profiles indicate a function for TCA substrate-specific BSH in mice administered with BPL-4 via gavage.

RT-qPCR was employed to assess the expression of signaling molecules linked to the FXR pathway in the liver and ileum tissues (Figure 6C,D). In comparison to the HCD group, the expression levels of FXR and short heterodimer partner (SHP) in the liver were markedly elevated in the HCD + BPL-4 group (*p* < 0.05), but the expression of FXR and its downstream genes, FGF15 and intestinal bile acid binding protein (IBABP), in the ileum was dramatically reduced (*p* < 0.05). CYP7A1 expression was markedly elevated in the livers of the HCD + BPL-4 group (*p* < 0.001), whereas sterol 27-hydroxylase (CYP27A1) and sterol 12α-hydroxylase (CYP8B1) expression was not significantly altered (*p* > 0.05). Given the negative feedback link between CYP7A1 and FXR, we posit that the cholesterol-lowering effect of BPL-4 is facilitated by reducing intestinal FXR, which in turn elevates CYP7A1 expression and diminishes endogenous cholesterol production.

## 4. Discussion

The global rise in living standards has popularized diets high in fat and calories, contributing to a growing incidence of dyslipidemia. This condition is a well-established risk factor for a spectrum of cardiovascular diseases, including hypertension, atherosclerosis, and coronary heart disease. Statins remain the first-line pharmaceutical therapy for hypercholesterolemia due to their efficacy. However, their associated side effects raise concerns regarding long-term use. Therefore, developing effective, safe, and well-tolerated alternative strategies for cholesterol management remains a major focus of biomedical research.

This study investigated the mechanisms by which *Lactobacillus*-derived BSH enzymes modulate cholesterol-lowering effects on microbial interactions. Building on prior investigations, the current investigation was conducted on mice using the gavage of BSH recombinant bacterium YB334, with subsequent macro-genome sequencing of their fecal samples to identify the ‘responsive’ strains of intestinal flora that exhibited substantial changes due to BSH exposure. Metagenomics analysis revealed substantial alterations in *B. pseudolongum* and *L. reuteri* compared with the empty-plasmid control. To further characterize these changes, mouse fecal samples were cultured on modified MRS agar plates enriched with lithium mupirocin and cysteine hydrochloride. After anaerobic incubation at 37 °C for 48 h, 66 colonies were isolated and identified as *Bifidobacterium* through biochemical methods and 16S rDNA sequencing. Among these, 54 colonies were classified as *B. pseudolongum*, representing 82% of the total, indicating significant advancement in subsequent experiments.

To elucidate the mechanism by which *B. pseudolongum* responds to BSH activity, we conducted in vitro assays by supplementing its culture medium with supernatants from recombinant YB334-BSH bacterial cultures. The findings indicated that the culture medium enriched with the YB334-BSH crude extract, which included the inducer, enhanced the proliferation of *B. pseudolongum* in a concentration-dependent manner. As the concentration increased from 5:1 to 10:1, the promoting effect on *B. pseudolongum* was amplified, with the BPL-4 strain exhibiting the most pronounced increase. The enhancement of *B. pseudolongum* growth by the YB334-BSH crude extract with the inducer was specific to the strain. It exerted no notable promotional effect on *B. adolescentis* BA-6 or *B. longum* BL-4 and BL-7, and even demonstrated inhibitory effects on *L. johnsonii* J5 and J16, as well as *L. reuteri* L2. Given that BSH enzymes catalyze the hydrolysis of TCA to CA, we examined the tolerance of various strains to CA and TCA. The findings indicated that several strains of *B. pseudolongum* and *B. adolescentis* BA-6 showed markedly greater tolerance to CA than to TCA, whereas *B. longum* BL-4 and BL-7 displayed significantly enhanced tolerance to TCA than to CA. *L. johnsonii* J5 and J16, together with *L. reuteri* L2, exhibited considerable tolerance to both TCA and CA. Taken together, these data suggest that under the influence of the YB334 BSH, TCA is hydrolyzed into CA, demonstrating significant tolerance to *B. pseudolongum* and thereby facilitating its proliferation. Certain bacterial strains in the gut microbiota exhibiting similar bile salt tolerance may also exhibit alterations in abundance due in response to BSH activity. These findings are consistent with previous studies. For example, Bulent Çetin et al. [33] demonstrated the probiotic potential of a raw milk-derived *E. faecium* strain, which showed robust BSH activity and bile tolerance. Likewise, Guangqiang Wang et al. [34] reported that BSH activity is a key determinant of bile salt tolerance in *Lactobacillus plantarum* AR113.

Subsequently, we conducted a murine experiment using *B. pseudolongum* BPL-4. The findings indicated that, in comparison with the control group, blood cholesterol levels, including TC, LDL-C, and HDL-C, were markedly diminished in high-cholesterol mice administered BPL-4. These results illustrate the cholesterol-reducing effect of BPL-4. An investigation into the mechanism of BPL-4’s cholesterol-lowering effects revealed that mice receiving BPL-4 through oral gavage demonstrated markedly decreased levels of conjugated bile acids and substantially elevated levels of unconjugated bile acids. Specifically, TCA levels were significantly diminished, whereas CA levels were significantly increased. In mouse livers, the expression of FXR and SHP was markedly elevated, whereas in the ileum, the expression of FXR and its downstream genes, FGF15 and IBABP, was dramatically diminished. Concurrently, hepatic CYP7A1 expression was notably increased. These results unequivocally indicated that the *B. pseudolongum* strain evaluated in this study, when administered to mice on a high-fat diet, significantly decreased FXR levels and augmented CYP7A1 enzymatic activity, thereby lowering cholesterol levels. This method of action aligns with the previously documented cholesterol-lowering mechanism of the YB334 recombinant strain through FXR [17]. FXR is a bile acid-activated transcription factor predominantly expressed in the liver and intestine. It primarily regulates bile acid homeostasis through a negative feedback mechanism by modulating the expression of CYP7A1, the rate-limiting enzyme in bile acid synthesis. This regulatory role is exemplified by several studies: Fei Li et al. [35] reported that the antioxidant tempol reduces gut Lactobacillus abundance and its bile salt hydrolase (BSH) activity, leading to the accumulation of the FXR antagonist tauro-β-muricholic acid (T-β-MCA). This inhibition of intestinal FXR signaling improved metabolic parameters in obesity. Similarly, Masaaki Miyata et al. [36] found that taurine supplementation alters ileal bile acid composition, increasing levels of FXR-antagonistic bile acids like T-β-MCA, which inhibits ileal FXR signaling and subsequently upregulates CYP7A1-mediated cholesterol catabolism. Furthermore, Minghua Yang et al. [37] demonstrated that cholesterol-lowering probiotics (e.g., Lactobacillus and Bifidobacterium) can modulate the FXR–FGF15 axis by altering gut microbiota composition and bile acid metabolism, thereby alleviating non-alcoholic fatty liver disease and dyslipidemia.

Our findings expand on prior reports on the probiotic effects of secondary metabolites produced by *B. pseudolongum*. Qian Song et al. [38] demonstrated that *B. pseudolongum* mitigates non-alcoholic fatty liver disease and hepatocarcinogenesis through the secretion of antitumor metabolites, such as acetate, via the gut–liver axis. Additionally, Ke Zhang et al. [39] indicated that carvacrol and thymol (CAT) enhances the abundance of *B. pseudolongum*, activates its cyclic guanosine monophosphate-protein kinase G pathway (cGMP-PKG) signaling pathway, and consequently inhibits dextran sulfate sodium-induced (DSS-induced) colitis. Nonetheless, metabolomic sequencing examination of BPL-4 samples fed via gavage to experimental mice in this investigation revealed no significant variations in the cholesterol-lowering effects of secondary metabolites produced by *B. pseudolongum*.

This study identified *B. pseudolongum* as a strain that notably responds to BSH, with its cholesterol-lowering mechanism mediated through the regulation of the bile acid receptor FXR signaling pathway. Importantly, the response of *B. pseudolongum* to BSH is characterized by markedly greater tolerance to the BSH catalytic product CA than to TCA. To our knowledge, this is the first study to establish a novel mechanism by which BSH modulates certain gut bacterial strains to reduce cholesterol levels. This study focused on the YB334 recombinant strain, which exhibits substrate specificity for TCA hydrolysis. However, BSH strains possess varying substrate specificities, including GCA and TβMCA. We hypothesized that variations in bile acid profiles resulting from BSH strains with distinct substrate specificities may have divergent effects on gut microbiota modulation [40]. Additional investigations are required to examine the impact of BSH strains with varying substrate specificities on the regulation of gut microbiota to enhance the understanding of how BSH lowers cholesterol levels through modulation of gut microbiota. These findings suggest that BSH-active lactic acid bacteria demonstrate significant potential for future development as a safe, effective, and dietary (food-grade) strategy for cholesterol management.

## 5. Conclusions

In this study, complementary in vivo and in vitro experiments were performed to comprehensively examine the mechanism of action of BSH activity in modulating gut microbiota and cholesterol metabolism. The research route and main results of this study are shown in Figure 7. This study demonstrates that the recombinant strain YB334, administered orally, significantly reduced cholesterol levels in high-cholesterol-diet mice and selectively enriched *B. pseudolongum* in the gut. Subsequent isolation and in vitro assays confirmed that BSH specifically promotes the growth of *B. pseudolongum* due to its high tolerance to CA, a BSH catalytic product. A particularly robust isolate, designated BPL-4, was shown to independently lower serum TC, LDL-C, and HDL-C levels in high-fat-diet mice. Mechanistic studies revealed that this effect is mediated through the bile acid receptor FXR signaling pathway. To the best of our knowledge, this is the first study to identify and characterize a ‘responsive strain’ within the gut microbiota that is promoted by BSH activity and that mediates the cholesterol-lowering effect of a probiotic. These findings provide a crucial theoretical foundation for developing novel probiotic-based therapies for hypercholesterolemia and have significant potential for application in the functional food and pharmaceutical industries.

## Figures and Tables

**Figure 1 nutrients-17-03019-f001:**
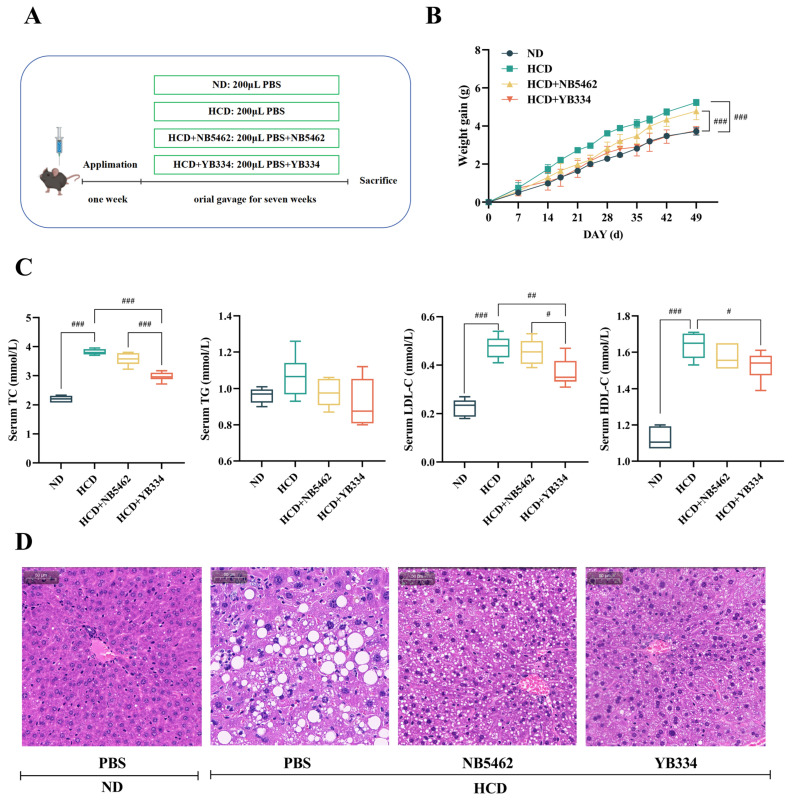
High-cholesterol modeling ((**A**): experimental design; (**B**): weight gain) and physiological and biochemical indexes ((**C**): serum lipid; (**D**): hematoxylin and eosin (H&E)-stained liver tissue section) were used to evaluate the fat-reducing function of BSH strain in these four groups (ND: normal diet; HCD: high-cholesterol diet; HCD + NB5462: high-cholesterol diet + empty plasmid control NB5462; HCD + YB334: high-cholesterol diet + YB334). TC: total cholesterol; TG: triglycerides; HDL-C: high-density lipoprotein cholesterol; LDL-C: low-density lipoprotein cholesterol. Values are expressed as mean ± SD (*n* = 6). Scale bars, 50 μm. White dots in H&E staining diagram indicate lipid droplets. # *p* < 0.05, ## *p* < 0.01, ### *p* < 0.001.

**Figure 2 nutrients-17-03019-f002:**
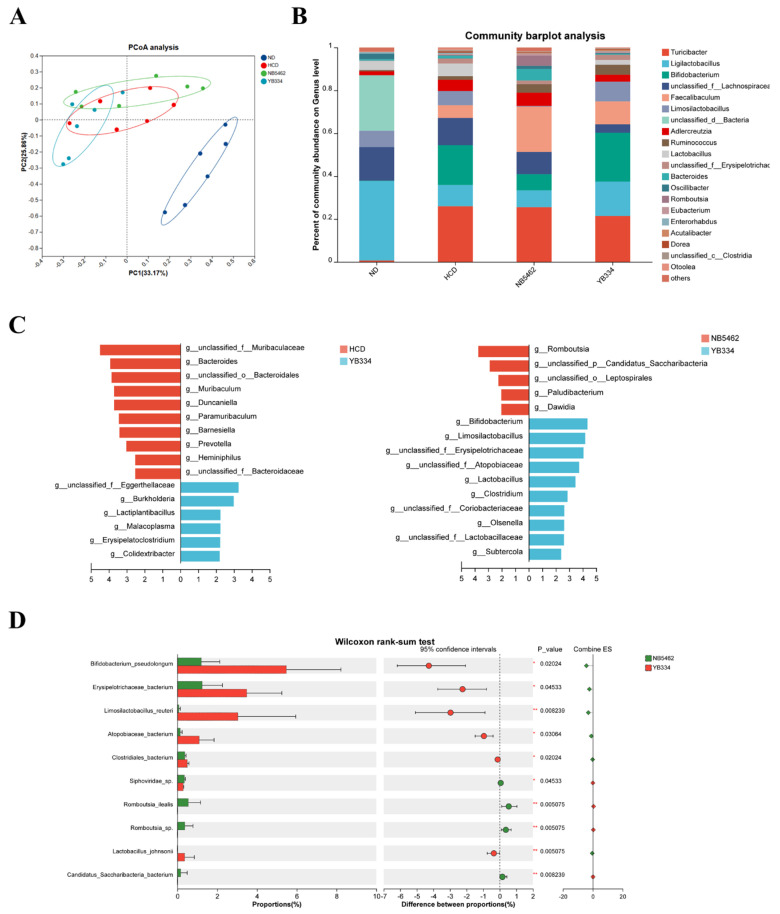
YB334 regulated the gut microbiota. (**A**) β-diversity analysis of sample metagenomes using principal coordinates analysis (PCoA). (**B**) Gut microbiota composition at species level. (**C**) Linear discriminant analysis effect size (LEfSe). (**D**) Differential species testing at the genus level. ND: normal diet; HCD: high-cholesterol diet; HCD + NB5462: high-cholesterol diet + empty plasmid control NB5462; HCD + YB334: high-cholesterol diet + YB334. Values are expressed as mean ± SD (*n* = 6).

**Figure 3 nutrients-17-03019-f003:**
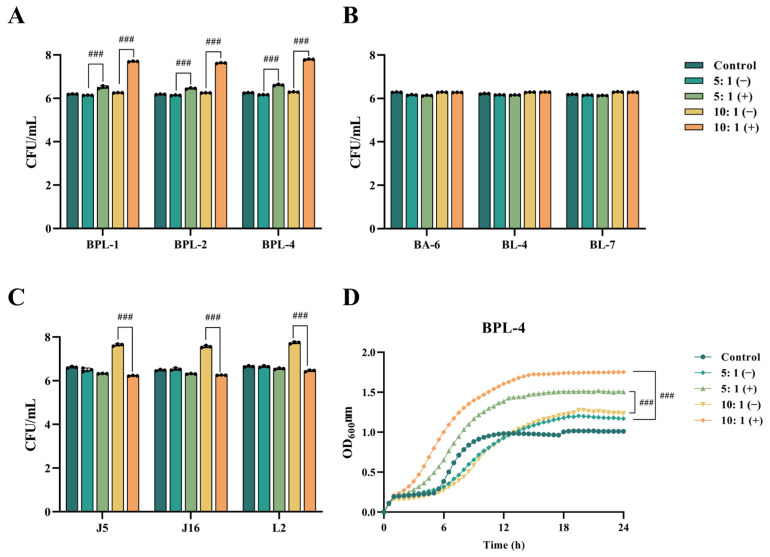
The effect of YB334-BSH enzyme crude extract on the growth of different strains. (**A**): *Bifidobacterium pseudolongum* BPL-1, BPL-2 and BPL-4. (**B**): *Bifidobacterium adolescentis* BA-6 and *Bifidobacterium longum* BL-4 and BL-7. (**C**): *Lactobacillus johnsonii* J5 and J16 and *Limosilactobacillus reuteri* L2. (**D**). BPL-4 growth curve. Values are expressed as mean ± SD (*n* = 3). ### *p* < 0.001.

**Figure 4 nutrients-17-03019-f004:**
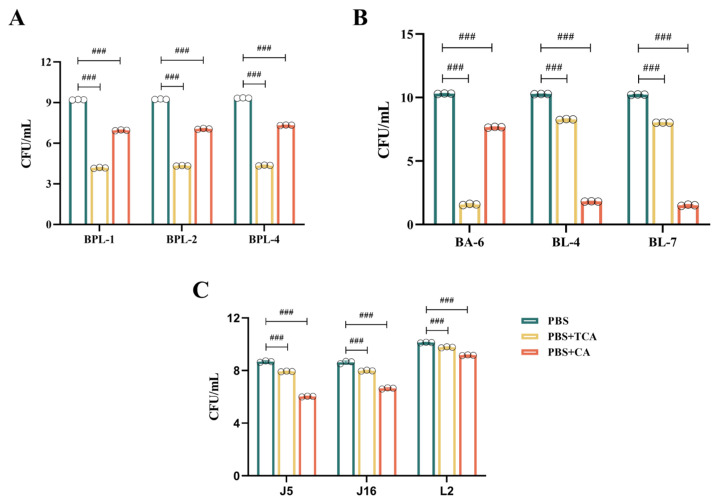
Growth of different strains when TCA and CA are added. (**A**): *B. pseudolongum* BPL-1, BPL-2 and BPL-4. (**B**): *B. adolescentis* BA-6 and *B. longum* BL-4 and BL-7. (**C**): *L. johnsonii* J5 and J16 and *L. reuteri* L2. Values are expressed as mean ± SD (*n* = 3). ### *p* < 0.001.

**Figure 5 nutrients-17-03019-f005:**
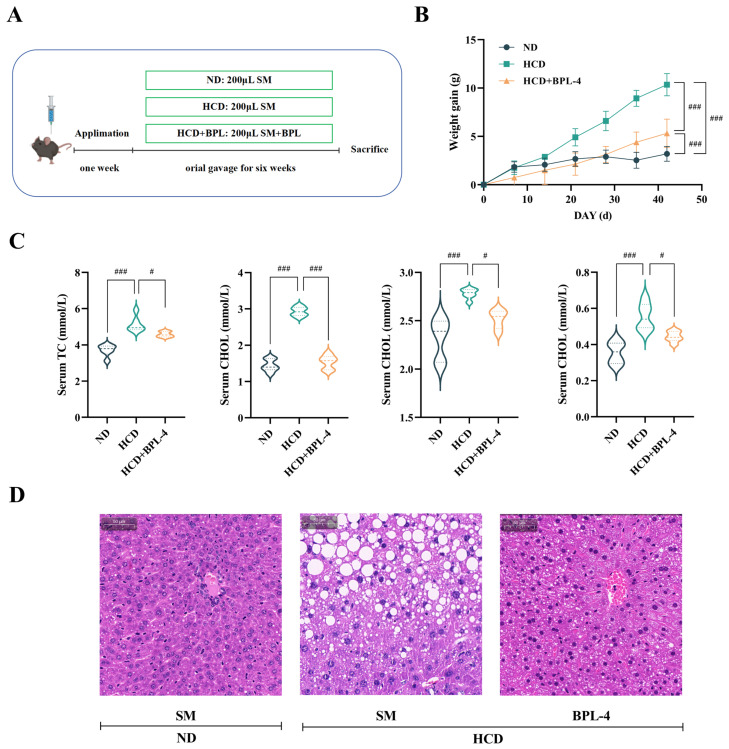
High-cholesterol modeling ((**A**): experimental design; (**B**): weight gain) and physiological and biochemical indexes ((**C**): serum lipid; (**D**): H&E stained liver tissue section) were used to evaluate the fat-reducing function of BSH strain in these three groups (ND: normal diet; HCD: high-cholesterol diet; HCD + BPL-4: high-cholesterol diet + BPL-4). TC: total cholesterol; TG: triglycerides; HDL-C: high-density lipoprotein cholesterol; LDL-C: low-density lipoprotein cholesterol. Values are expressed as mean ± SD (*n* = 6). Scale bars, 50 μm. White dots in H&E staining diagram indicate lipid droplets. # *p* < 0.05, ### *p* < 0.001.

**Figure 6 nutrients-17-03019-f006:**
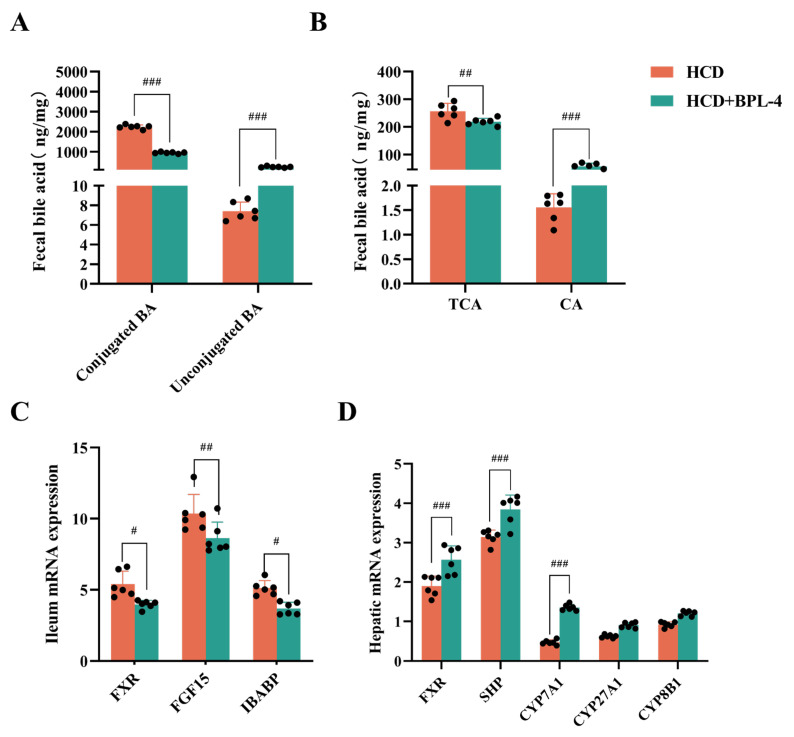
Fecal bile acid ((**A**): conjugated and unconjugated BA; (**B**): TCA and CA) and genes related to hepatic FXR (**C**) and ileal FXR (**D**) of SPF mice were used to analyze the TC-reducing metabolism of BSH. HCD: high-cholesterol diet; HCD + BPL-4: high-cholesterol diet + BPL-4. TCA: taurocholic acid; CA: cholic acid. Values are expressed as mean ± SD (*n* = 6). # *p* < 0.05, ## *p* < 0.01, ### *p* < 0.001.

**Figure 7 nutrients-17-03019-f007:**
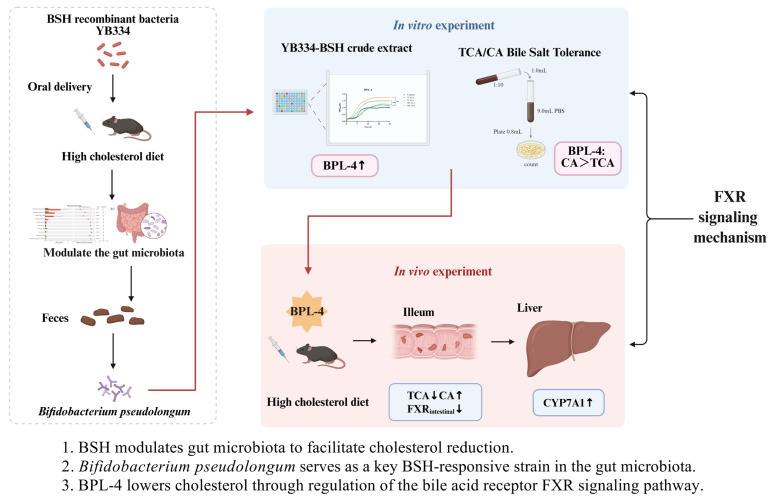
Flowchart of the experimental design.

**Table 1 nutrients-17-03019-t001:** Bacterial strains used in this study.

Strains	Descriptions ^a^	Source or Reference
*Lactobacillus johnsonii*		
YH334	Wide type	Laboratory screening [17]
J5	Wide type	This study
J16	Wide type	This study
*Lactiplantibacillus plantarum*		
YB334	Em^r^, the pSIP334 plasmid was introduced into strain WCFS1Δ*bsh*	Laboratory construction [17]
NB5462	Em^r^, the pSIP334 plasmid was introduced into strain WCFS1Δ*bsh*	Laboratory construction [17]
*Bifidobacterium pseudolongum*		
BPL	Wide type	This study
*Bifidobacterium adolescentis*		
BA-6	Wide type	This study
*Bifidobacterium longum*		
BL-4	Wide type	This study
BL-7	Wide type	This study
*Limosilactobacillus reuteri*		
L2	Wide type	This study

^a^ Em^r^, Erythromycin resistance.

**Table 2 nutrients-17-03019-t002:** Primer sequence in this study.

Primer	Sequence (5′-3′)	Reference
FXR-F	GGAACTCCGGACATTCAAC	[24]
FXR-R	GTGTCCATCACTGCACATC	
SHP-F	TCCTAGCCAAGACAGTAGCCTTCC	[25]
SHP-R	TACCGCTGCTGGCTTCCTCTAG	
CYP7A1-F	GCTAAGACGCACCTCGTGATCC	
CYP7A1-R	CCGCAGAGCCTCCTTGATGATG	
FGF15-F	CGGTCGCTCTGAAGACGATTGC	
FGF15-R	TACATCCTCCACCATCCTGAACGG	
CYP27A1-F	ATTAAGGAGACCCTGCGCCT	
CYP27A1-R	AGGCAAGACCGAACCCCATA	
CYP8B1-F	AAGGCTGGCTTCCTGAGCTT	[26]
CYP8B1-R	AACAGCTCATCGGCCTCATC	
IBABP-F	GGCCCGCAACTTCAAGATC	[27]
IBABP-R	TAGTGCTGGGACCAAGTGAAGTC	
rpL32-F	TCTGGTCCACAACGTCAAGG	[17]
rpL32-R	GGATTGGTGACTCTGATGGC	

**Table 3 nutrients-17-03019-t003:** BSH activities of different *Bifidobacterium pseudolongum* strains.

Strain	Enzyme Activity (U/mL)	
	GCA	TCA
BPL-1	0.84 ± 0.06 ***	1.83 ± 0.07 ***
BPL-2	0.44 ± 0.03 ***	1.37 ± 0.03 ***
BPL-4	2.11 ± 0.05 ***	2.53 ± 0.04 ***
BPL-5	0.84 ± 0.02 ***	1.93 ± 0.07 ***
BPL-8	0.95 ± 0.11 ***	1.90 ± 0.07 ***
BPL-9	0.93 ± 0.02 ***	1.60 ± 0.03 ***
BPL-10	1.24 ± 0.04 ***	2.96 ± 0.08 ***
BPL-11	0.60 ± 0.08 ***	1.71 ± 0.07 ***
NB5462	0.07 ± 0.01	0.07 ± 0.02

Values are expressed as mean standard deviation (*n* = 3). *** Represents a significant different from NB5462 (*p* < 0.001).

## Data Availability

Data can be accessed from the corresponding author upon request.

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
