# Peer review of "Cholesterol-Lowering Mechanism of Lactobacillus Bile Salt Hydrolase Through Regulation of Bifidobacterium pseudolongum in the Gut Microbiota"

_nutrients, 2025, doi:10.3390/nu17183019_

Round 1

Reviewer 1 Report

Comments and Suggestions for Authors

After some revisions, I believe this manuscript can be considered for publication in the Nutrients journal. These are my suggestions:

The abstract should be better organized and structured. The main objectives are not clear; what were the applied methodologies and the main results (values with statistical significance)? Future perspectives should be stated at the end of the conclusions.

The Introduction is too brief. Please, provide more data on the addressed topics; the study’s rationale and novelty are not elucidated.

Ethics: Approval date for the study is not mentioned.

Please, provide more details regarding the liver histological analysis and the metagenomics analysis.

The inclusion of a flowchart or a graphical abstract with all the steps taken in the research would be helpful for the readers.

The quality and size of figure 2 have to be improved.

 The Results are well presented, but the Discussion section has to be improved. More data from other studies should be further discussed, and it could be better organized in subsections, aligned with the Results.

Further discussions on the possible extrapolation for humans should be given.

The Conclusions should be more succinct, and future perspectives need to be elaborated.

Author Response

Comments 1: The abstract should be better organized and structured. The main objectives are not clear; what were the applied methodologies and the main results (values with statistical significance)? Future perspectives should be stated at the end of the conclusions.

Response 1:

Thank you for pointing this out. We agree with this comment. Therefore, we have now thoroughly revised the abstract to improve its structure and clarity. The updated text is provided below (Page 1, Section Abstract, Line 12-35):

Background: Cardiovascular diseases (CVDs) represent a major global health burden, and cholesterol reduction is a key strategy for their prevention and management. This study investigated the mechanism by which bile salt hydrolase (BSH) from Lactobacilli reduces cholesterol levels by modulating the growth of Bifidobacterium pseudolongum. Methods: The BSH-recombinant strain YB334 was administered to high-cholesterol diet mice and the cholesterol-lowering function of the strain was evaluated by assessing serum cholesterol parameters, including total cholesterol (TC), low-density lipoprotein (LDL) and high-density lipoprotein (HDL). Metagenomic sequencing was used to analyze the gut microbiota, leading to the screening and acquisition of the "responsive" strains affected by BSH. Subsequent investigations were conducted into their cholesterol-lowering effects and mechanisms of action. Results: Oral administration of the BSH-recombinant strain YB334 can effectively reduced serum cholesterol levels in hypercholesterolemic mice, while simultaneously leading to a significant increase in the abundance of B. pseudolongum within the gut microbiota. In vitro experiments indicated that this increased abundance might be closely associated with the strain's high tolerance to CA, the catalytic product of the BSH enzyme. The BPL-4 strain, obtained through screening, demonstrated cholesterol-lowering efficacy. Mechanistically, BPL-4 altered bile acid pool composition and modulated the farnesoid X receptor (FXR) signaling axis: it suppressed ileal FXR-fibroblast growth factor 15 (FGF15) expression, thereby de-repressing hepatic cholesterol 7α-hydroxylase (CYP7A1) and accelerating cholesterol catabolism into bile acids. Conclusions: This study provides the first evidence that BSH from lactobacilli can shape the signature gut microbiota by modulating bile acid metabolism via the FXR-CYP7A1 axis, thereby demonstrating a mechanism for cholesterol-lowering.”

Comments 2: The Introduction is too brief. Please, provide more data on the addressed topics; the study’s rationale and novelty are not elucidated.

Response 2:

Thank you for pointing this out. We agree with this comment. Therefore, we have revised the Introduction section to provide more background on the addressed topics. The updated text is provided below (Page 2-3, Section 1, Line 86-91):

Previous studies from our laboratory have shown that when the BSH recombinant strain YB81 was administered to PGF mice with antibiotic-depleted gut microbiota, this strain did not exhibit cholesterol-lowering effects. However, in SPF mouse models with normal gut microbiota, the same strain demonstrated significant cholesterol-lowering effects. These results indicate that the gut microbiota directly contributes to the cholesterol-lowering effects of certain BSH from.from Lactobacilli [20].

Additionally, we have clarified the theoretical basis and distinctive features of the research. The updated text is provided below (Page 3, Section 1, Line 91-96):

Building on these findings, this study aims to elucidate the cholesterol-lowering mechanisms of the recombinant strain YB334 by analyzing BSH enzyme alterations in the structure and function of the gut microbiota. We seek to uncover novel mechanisms of cholesterol regulation by lactic acid bacterial BSH from the perspective of host-microbe interactions, thereby providing a theoretical foundation for developing probiotic therapies against hypercholesterolemia.

Comments 3: Ethics: Approval date for the study is not mentioned.

Response 3:

Thank you for pointing this out. We agree with this comment. Therefore, we have supplemented the approval date in the Methods section. The revised sentence now reads (Page 3, Section 2.2, Line 107):

“The animal study protocol was approved by the Animal Ethics Committee of Nanjing Normal University (Approval Number: SYXK 2020-0047; Approval Date: 15 February 2025), and all experiments were performed in accordance with relevant guidelines and regulations.”

Comments 4: Please, provide more details regarding the liver histological analysis and the metagenomics analysis.

Response 4:

Thank you for pointing this out. We agree with this comment. Therefore, we have now added more detailed descriptions of the liver histological analysis methods and expanded our discussion of these findings in the Conclusions section. The updated text is provided below.

Methods (Page 4, Section 2.4, Line 154-159):

Liver tissues were fixed in 4% paraformaldehyde for 24 h, embedded in paraffin, and sectioned at a thickness of 4 μm. Sections were stained with hematoxylin and eosin (H&E) and imaged using an upright optical microscope (Nikon Eclipse E100, Japan). All histological procedures were performed by Wuhan Servicebio Technology Co., Ltd (Wuhan, China).

Conclusions 1 (Page 7, Section 3.1, Line 277-283):

“Figure 1D shows a histological study of the liver tissue stained with H&E. The HCD group displayed substantial microvesicular steatosis, characterized by abundant lipid droplet accumulation, which was absent in the ND group. All intervention groups exhibited varying degrees of attenuation in hepatic lipid deposition and vacuolation. The most significant amelioration was observed in the HCD + YB334 group, where hepatocyte morphology was largely restored to normal, resulting in superior outcomes compared to the HCD + NB5462 group.

Conclusions 2 (Page 12, Section 3.6, Line 428-432):

“Figure 5D shows a histological study of the liver tissue stained with H&E. Compared to the HCD + BPL-4 group, mice on the HCD alone exhibited substantial lipid droplet accumulation and marked fatty degeneration in adipose tissue, pathologies that were absent in the ND group. This indicates that BPL-4 effectively alleviates HCD-induced hepatic steatosis.

Additionally, we have now added more detailed descriptions of the metagenomics analysis methodology and expanded the discussion of these results in the Conclusions section. The updated text is provided below.

Methods (Page 4-5, Section 2.5, Line 161-178):

” Metagenomic DNA was extracted from mouse feces using the E.Z.N.A.® Soil DNA Kit (Omega Bio-tek, USA). Sequencing was performed on an Illumina NovaSeq/HiSeq Xten platform (Illumina, USA) following bridge PCR amplification. The remaining high-quality reads were de novo assembled using MEGAHIT, with contigs ≥ 300 bp retained for subsequent analysis. Non-redundant gene catalogs were constructed from the assembled contigs using MetaGene for prediction and CD-HIT for clustering. All laboratory procedures and sequencing were conducted by Shanghai Majorbio Bio-Pharm Technology Co., Ltd. (China), and the bioinformatic analysis was performed on the Majorbio Cloud Platform (https://cloud.majorbio.com).

To identify microbial taxa with significant abundance differences between groups, linear discriminant analysis effect size (LEfSe) was employed, with a linear discriminant analysis (LDA) score threshold of > 2. A custom reference database of bile salt hydrolase (BSH) genes was compiled by querying the UniProt protein database. Beta-diversity between experimental groups was assessed via principal coordinates analysis (PCoA) based on Bray-Curtis distances. Statistical significance of inter-group differences was determined using the Wilcoxon rank-sum test, with p-values adjusted for false discovery rate (FDR).

Conclusions (Page 9, Section 3.2, Line 326-329):

These findings suggest that the cholesterol-lowering effect of the recombinant strain YB334 is mediated through the enrichment of specific beneficial bacteria, notably B. pseudolongum, highlighting this bacterium as a key microbial target for elucidating the underlying mechanism.

Comments 5: The inclusion of a flowchart or a graphical abstract with all the steps taken in the research would be helpful for the readers.

Response 5:

Thank you for pointing this out. We agree with this comment. Therefore, we have now included a clear flowchart outlining the research steps. This figure is provided below for the reviewer's convenience (Page 17, Section 5, Line 588-589).

Figure 7. Flow chart of the experimental design.(The figures are included in the document.)

Comments 6: The quality and size of figure 2 have to be improved.

Response 6:

Thank you for pointing this out. We agree with this comment. Therefore, as the image in Figure 2 is rather complex, we have submitted more distinct individual images.

Comments 7: The Results are well presented, but the Discussion section has to be improved. More data from other studies should be further discussed, and it could be better organized in subsections, aligned with the Results.

Response 7:

Thank you for pointing this out. We agree with this comment. We have thoroughly revised the Discussion section to provide a more in-depth interpretation of our findings and to better relate them to existing literature. The updated text is provided below (Page 14-16, Section 4, Line 468-570):

The global rise in living standards has popularized diets high in fat and calories, contributing to a growing incidence of dyslipidemia. This condition is a well-established risk factor for a spectrum of cardiovascular diseases, including hypertension, atherosclerosis, and coronary heart disease. Statins remain the first-line pharmaceutical therapy for hypercholesterolemia due to their efficacy. However, their associated side effects raise concerns regarding long-term use. Therefore, developing effective, safe, and well-tolerated alternative strategies for cholesterol management remains a major focus of biomedical research.

This study investigated the mechanisms by which Lactobacillus-derived BSH enzymes modulate cholesterol-lowering effects on microbial interactions. Building on prior investigations, the current investigation was conducted on mice using the gavage of BSH recombinant bacterium YB334, with subsequent macro-genome sequencing of their fecal samples to identify the 'responsive' strains of intestinal flora that exhibited substantial changes due to BSH exposure. Metagenomics analysis revealed substantial alterations in B. pseudolongum and L. reuteri compared with the empty-plasmid control. To further characterize these changes, mouse fecal samples were cultured on modified MRS agar plate enriched with lithium mupirocin and cysteine hydrochloride. After anaerobic incubation at 37 °C for 48 h, 66 colonies were isolated and identified as Bifidobacterium through biochemical methods and 16S rDNA sequencing. Among these, 54 colonies were classified as B. pseudolongum, representing 82% of the total, indicating significant advancement in subsequent experiments.

To elucidate the mechanism by which B. pseudolongum responds to BSH activity, we conducted in vitro assays by supplementing its culture medium with supernatants from recombinant YB334-BSH bacterial cultures. The findings indicated that the culture medium enriched with the YB334-BSH crude extract, which included the inducer, enhanced the proliferation of B. pseudolongum in a concentration-dependent manner. As the concentration increased from 5:1 to 10:1, the promoting effect on B. pseudolongum was amplified, with the BPL-4 strain exhibiting the most pronounced increase. The enhancement of B. pseudolongum growth by the YB334-BSH crude extract with the inducer was specific to the strain. It exerted no notable promotional effect on B. adolescentis BA-6 and B. longum BL-4, and BL-7, and even demonstrated inhibitory effects on L. johnsonii J5 and J16, as well as L. reuteri L2. Given that BSH enzymes catalyze the hydrolysis of TCA to CA, we examined the tolerance of various strains to CA and TCA. The findings indicated that several strains of B. pseudolongum and B. adolescentis BA-6 showed markedly greater tolerance to CA than to TCA, whereas B. longum BL-4 and BL-7 displayed significantly enhanced tolerance to TCA than to CA. L. johnsonii J5 and J16, together with L. reuteri L2, exhibited considerable tolerance to both TCA and CA. Taken together, these data suggest that under the influence of the YB334 BSH, TCA is hydrolyzed into CA, demonstrating significant tolerance to B. pseudolongum and thereby facilitating its proliferation. Certain bacterial strains in the gut microbiota exhibiting similar bile salt tolerance may also exhibit alterations in abundance due in response to BSH activity. These findings are consistent with previous studies. For example, Bulent Çetin et.al [33] demonstrated the probiotic potential of a raw milk-derived E. faecium strain, which showed robust BSH activity and bile tolerance. Likewise, Guangqiang Wang et al. [34] reported that BSH activity is a key determinant of bile salt tolerance in Lactobacillus plantarum AR113.

Subsequently, we conducted a murine experiment using B. pseudolongum BPL-4. The findings indicated that, in comparison with the control group, blood cholesterol levels, including TC, LDL-C, and HDL-C, were markedly diminished in high-cholesterol mice administered BPL-4. These results illustrate the cholesterol-reducing effect of BPL-4. An investigation into the mechanism of BPL-4's cholesterol-lowering effects revealed that mice receiving BPL-4 through oral gavage demonstrated markedly decreased levels of conjugated bile acids and substantially elevated levels of unconjugated bile acids. Specifically, TCA levels were significantly diminished, whereas CA levels were significantly increased. In mouse livers, the expression of FXR and SHP was markedly elevated, whereas in the ileum, the expression of FXR and its downstream genes, FGF15 and IBABP, was dramatically diminished. Concurrently, hepatic CYP7A1 expression was notably increased. These results unequivocally indicated that the B. pseudolongum strain evaluated in this study, when administered to mice on a high-fat diet, significantly decreased FXR levels and augmented CYP7A1 enzymatic activity, thereby lowering cholesterol levels. This method of action aligns with the previously documented cholesterol-lowering mechanism of the YB334 recombinant strain through FXR [17]. FXR is a bile acid-activated transcription factor predominantly expressed in the liver and intestine. It primarily regulates bile acid homeostasis through a negative feedback mechanism by modulating the expression of CYP7A1, the rate-limiting enzyme in bile acid synthesis. This regulatory role is exemplified by several studies: Fei Li et al. [35] reported that the antioxidant tempol reduces gut Lactobacillus abundance and its bile salt hydrolase (BSH) activity, leading to the accumulation of the FXR antagonist tauro-β-muricholic acid (T-β-MCA). This inhibition of intestinal FXR signaling improved metabolic parameters in obesity. Similarly, Masaaki Miyata et al. [36] found that taurine supplementation alters ileal bile acid composition, increasing levels of FXR-antagonistic bile acids like T-β-MCA, which inhibits ileal FXR signaling and subsequently upregulates CYP7A1-mediated cholesterol catabolism. Furthermore, Minghua Yang et al. [37] demonstrated that cholesterol-lowering probiotics (e.g., Lactobacillus and Bifidobacterium) can modulate the FXR–FGF15 axis by altering gut microbiota composition and bile acid metabolism, thereby alleviating non-alcoholic fatty liver disease and dyslipidemia.

Our findings expand on prior reports on the probiotic effects of secondary metabolites produced by B. pseudolongum. Qian Song et al. [38] demonstrated that B. pseudolongum mitigates non-alcoholic fatty liver disease and hepatocarcinogenesis through the secretion of antitumor metabolites, such as acetate, via the gut-liver axis. Additionally, Ke Zhang et al. [39] indicated that carvacrol and thymol (CAT) enhances the abundance of B. pseudolongum, activates its cyclic guanosine monophosphate-protein kinase G pathway (cGMP-PKG) signalling pathway, and consequently inhibits dextran sulfate sodium-induced (DSS-induced) colitis. Nonetheless, metabolomic sequencing examination of BPL-4 samples fed via gavage to experimental mice in this investigation revealed no significant variations in the cholesterol-lowering effects of secondary metabolites produced by B. pseudolongum (results not shown).

This study identified B. pseudolongum as a strain that notably responds to BSH, with its cholesterol-lowering mechanism mediated through the regulation of the bile acid receptor FXR signaling pathway. Importantly, the response of B. pseudolongum to BSH is characterized by markedly greater tolerance to the BSH catalytic product CA than to TCA. To our knowledge, this is the first study to establish a novel mechanism by which BSH modulates certain gut bacterial strains to reduce cholesterol levels. This study focused on the YB334 recombinant strain, which exhibits substrate specificity for TCA hydrolysis. However, BSH strains possess varying substrate specificities, including GCA and TβMCA. We hypothesized that variations in bile acid profiles resulting from BSH strains with distinct substrate specificities may have divergent effects on gut microbiota modulation [40]. Additional investigations are required to examine the impact of BSH strains with varying substrate specificities on the regulation of gut microbiota to enhance the understanding of how BSH lowers cholesterol levels through modulation of gut microbiota. These findings suggest that BSH-active lactic acid bacteria demonstrate significant potential for future development as a safe, effective, and dietary (food-grade) strategy for cholesterol management.

Comments 8: Further discussions on the possible extrapolation for humans should be given.

Response 8:

Thank you for pointing this out. We agree with this comment. We have now revised the Discussion section to place a stronger emphasis on the importance of cholesterol reduction for human health. Furthermore, we have expanded the discussion on the theoretical and practical prospects for developing cholesterol-lowering probiotics as a safe and effective novel therapeutic strategy. These revisions can be found primarily in the first and final paragraphs of the Discussion section.

First paragraph (Page 14, Section 4, Line 469-476):

The global rise in living standards has popularized diets high in fat and calories, contributing to a growing incidence of dyslipidemia. This condition is a well-established risk factor for a spectrum of cardiovascular diseases, including hypertension, atherosclerosis, and coronary heart disease. Statins remain the first-line pharmaceutical therapy for hypercholesterolemia due to their efficacy. However, their associated side effects raise concerns regarding long-term use. Therefore, developing effective, safe, and well-tolerated alternative strategies for cholesterol management remains a major focus of biomedical research.

Final paragraph (Page 16, Section 4, Line 555-570):

This study identified B. pseudolongum as a strain that notably responds to BSH, with its cholesterol-lowering mechanism mediated through the regulation of the bile acid receptor FXR signaling pathway. Importantly, the response of B. pseudolongum to BSH is characterized by markedly greater tolerance to the BSH catalytic product CA than to TCA. To our knowledge, this is the first study to establish a novel mechanism by which BSH modulates certain gut bacterial strains to reduce cholesterol levels. This study focused on the YB334 recombinant strain, which exhibits substrate specificity for TCA hydrolysis. However, BSH strains possess varying substrate specificities, including GCA and TβMCA. We hypothesized that variations in bile acid profiles resulting from BSH strains with distinct substrate specificities may have divergent effects on gut microbiota modulation [40]. Additional investigations are required to examine the impact of BSH strains with varying substrate specificities on the regulation of gut microbiota to enhance the understanding of how BSH lowers cholesterol levels through modulation of gut microbiota. These findings suggest that BSH-active lactic acid bacteria demonstrate significant potential for future development as a safe, effective, and dietary (food-grade) strategy for cholesterol management.

Comments 9: The Conclusions should be more succinct, and future perspectives need to be elaborated.

Response 9:

Thank you for pointing this out. We agree with this comment. We have streamlined the Conclusions to make them more succinct and have added a new paragraph to elaborate on future research perspectives. The updated text is provided below (Page 16-17, Section 5, Line 571-578):

“In this study, complementary in vivo and in vitro experiments were performed to comprehensively examine the mechanism of action of BSH activity in modulating gut microbiota and cholesterol metabolism. The research route and main results of this study are shown in Figure 7. This study demonstrates that the recombinant strain YB334, administered orally, significantly reduced cholesterol levels in high-cholesterol-diet mice and selectively enriched B. pseudolongum in the gut. Subsequent isolation and in vitro assays confirmed that BSH specifically promotes the growth of B. pseudolongum due to its high tolerance to CA, a BSH catalytic product. A particularly robust isolate, designated BPL-4, was shown to independently lower serum TC, LDL-C, and HDL-C levels in high-fat-diet mice. Mechanistic studies revealed that this effect is mediated through the bile acid receptor FXR signaling pathway. To our knowledge, this is the first study to identify and characterize a 'responsive strain' within the gut microbiota that is promoted by BSH activity and that mediates the cholesterol-lowering effect of a probiotic. These findings provide a crucial theoretical foundation for developing novel probiotic-based therapies for hypercholesterolemia and have significant potential for application in the functional food and pharmaceutical industries.

Reviewer 2 Report

Comments and Suggestions for Authors

The manuscript presents an interesting topic; however, several critical details are missing or insufficiently described, which significantly affects the clarity and reproducibility of the work.

First, the abstract is difficult to follow as it is filled with abbreviations that are not expanded upon at first mention. This makes it confusing for readers who are not already familiar with the specific terminology. Terms such as “inducer” and “extract of X” should be clearly defined.

The methods section is also incomplete. The number of animals used in the study is not mentioned, making it impossible to assess the statistical power or validity of the findings. Additionally, the manuscript does not clearly explain how cholesterol levels were measured, nor does it provide sufficient details about the extraction protocol and other experimental parameters.

The metagenomic analysis is presented in a way that is not visually clear — the graphs are difficult to interpret, and proper legends and axis labels should be included. Furthermore, the rationale for growing Lactobacillus strains under anaerobic conditions should be justified, particularly if these strains are naturally capable of growing under aerobic conditions.

Finally, the study focuses exclusively on cholesterol levels when exploring cardiovascular disease risk. Incorporating proteomics or transcriptomics data could have provided a much deeper understanding of the underlying molecular mechanisms and may have strengthened the overall impact of the study.

Author Response

Comments 1: First, the abstract is difficult to follow as it is filled with abbreviations that are not expanded upon at first mention. This makes it confusing for readers who are not already familiar with the specific terminology. Terms such as “inducer” and “extract of X” should be clearly defined.

Response 1:

Thank you for pointing this out. We agree with this comment. We have now thoroughly revised the abstract to address this issue. The revised sentence now reads (Page 1, Section Abstract, Line 12-35):

Background: Cardiovascular diseases (CVDs) represent a major global health burden, and cholesterol reduction is a key strategy for their prevention and management. This study investigated the mechanism by which bile salt hydrolase (BSH) from Lactobacilli reduces cholesterol levels by modulating the growth of Bifidobacterium pseudolongum. Methods: The BSH-recombinant strain YB334 was administered to high-cholesterol diet mice and the cholesterol-lowering function of the strain was evaluated by assessing serum cholesterol parameters, including total cholesterol (TC), low-density lipoprotein (LDL) and high-density lipoprotein (HDL). Metagenomic sequencing was used to analyze the gut microbiota, leading to the screening and acquisition of the "responsive" strains affected by BSH. Subsequent investigations were conducted into their cholesterol-lowering effects and mechanisms of action. Results: Oral administration of the BSH-recombinant strain YB334 can effectively reduced serum cholesterol levels in hypercholesterolemic mice, while simultaneously leading to a significant increase in the abundance of B. pseudolongum within the gut microbiota. In vitro experiments indicated that this increased abundance might be closely associated with the strain's high tolerance to CA, the catalytic product of the BSH enzyme. The BPL-4 strain, obtained through screening, demonstrated cholesterol-lowering efficacy. Mechanistically, BPL-4 altered bile acid pool composition and modulated the farnesoid X receptor (FXR) signaling axis: it suppressed ileal FXR-fibroblast growth factor 15 (FGF15) expression, thereby de-repressing hepatic cholesterol 7α-hydroxylase (CYP7A1) and accelerating cholesterol catabolism into bile acids. Conclusions: This study provides the first evidence that BSH from lactobacilli can shape the signature gut microbiota by modulating bile acid metabolism via the FXR-CYP7A1 axis, thereby demonstrating a mechanism for cholesterol-lowering.”

Comments 2: The methods section is also incomplete. The number of animals used in the study is not mentioned, making it impossible to assess the statistical power or validity of the findings. Additionally, the manuscript does not clearly explain how cholesterol levels were measured, nor does it provide sufficient details about the extraction protocol and other experimental parameters.

Response 2:

Thank you for pointing this out. We agree with this comment. Therefore, we have supplemented the number of animals. Moreover, we have included the animal count (n=6) in the captions of the corresponding figures. The revised sentence now reads (Page 4, Section 2.2, Line 117):

“Following one week of adaptive feeding, the mice were divided into two groups: the control group received a ND, while the experimental group was fed a HCD to induce hypercholesterolaemia. Six mice were allocated to each group.

Additionally, cholesterol levels are determined by measuring the concentrations of total cholesterol (TC), triglycerides (TG), high-density lipoprotein cholesterol (HDL-C), and low-density lipoprotein cholesterol (LDL-C) in serum. We apologize for the lack of clarity in our original manuscript. Sampling and operational procedures are detailed in Section 2.3 (Serum biochemical parameter analysis; Page 4, Section 2.3, Line 146-152):

“Whole blood was collected from mice via retro-orbital bleeding. After incubation at room temperature for 30 min, the samples were centrifuged at 1500 ×g for 15 min to isolate serum. The serum levels of total cholesterol (TC), triglycerides (TG), high-density lipoprotein cholesterol (HDL-C) and low-density lipoprotein cholesterol (LDL-C) were quantified using an automated biochemistry analyzer (Hitachi 7100) at the Animal Centre of Nanjing Medical University.”

Comments 3: The metagenomic analysis is presented in a way that is not visually clear — the graphs are difficult to interpret, and proper legends and axis labels should be included. Furthermore, the rationale for growing Lactobacillus strains under anaerobic conditions should be justified, particularly if these strains are naturally capable of growing under aerobic conditions.

Response 3:

Thank you for pointing this out. We agree with this comment. We conducted metagenomic analysis on fecal samples from the experimental group mice, as referenced in literature:

[1] Feng, L.; Guo, Z.; Yao, W.; Mu, G.; Zhu, X. Metagenomics and Untargeted Metabolomics Analysis Revealed the Probiotic and Postbiotic Derived from Lactiplantibacillus plantarum DPUL F232 Alleviate Whey Protein-Induced Food Allergy by Reshaping Gut Microbiota and Regulating Key Metabolites. J Agric Food Chem 2024, 72, 25436-25448.

We selected and presented key findings in Figures 2A-D.

Figure 2A: The principal coordinates analysis (PCoA) plot visualizes data through dimensionality reduction, directly reflecting the degree of sample reproducibility within groups and the magnitude of differences between groups.

Figure 2B: The bar plot of microbial communities statistically summarizes the abundance of species, genes, or functions in each sample at taxonomic, genetic, or functional levels. It intuitively visualizes the composition of dominant species, genes, or functions in the community through a bar chart representation.

Figure 2C: The linear discriminant analysis effect size (LEfSe) plot is used to identify species features that best explain differences between two or more groups under different biological or environmental conditions, as well as the extent of their contribution to intergroup variation. The horizontal coordinate represents the LDA score obtained through linear discriminant analysis (LDA), where a higher LDA score indicates a greater influence of species abundance on the observed differential effects.

Figure 2D: Wilcoxon rank-sum test was applied to examine differences in genus-level abundances between different groups. The horizontal axis represents the percentage value of the abundance of a specific species in the sample.

In the revised manuscript, we have adjusted the figure dimensions to enhance clarity and readability.

Additionally, we have now supplemented the Materials and Methods section with information regarding the oxygen requirements of the experimental strains to justify the culture conditions. The revised sentence now reads (Page 3, Section 2.1, Line 95-98):

Lactobacillus and Limosilactobacillus were cultured aerobically in de Man, Rogosa and Sharpe (MRS) medium, while Bifidobacterium was cultured anaerobically in modified MRS agar medium (supplemented with mupirocin lithium salt and cysteine hydrochloride).”

(Page 10, Section 3.3, Line 351-352):

In parallel, under aerobic conditions, Lactobacillus johnsonii J5 and J16 and L. reuteri L2 were isolated.

Comments 4: Finally, the study focuses exclusively on cholesterol levels when exploring cardiovascular disease risk. Incorporating proteomics or transcriptomics data could have provided a much deeper understanding of the underlying molecular mechanisms and may have strengthened the overall impact of the study.

Response 4:

Thank you for pointing this out. We agree with this comment. To further investigate the mechanism by which BSH reduces cholesterol levels via Bifidobacterium pseudolongum BPL-4, we will employ quantitative bile acid analysis and non-targeted metabolomics on experimental samples in subsequent experiments.